# Effect of Replacing Dietary Corn with Broken Rice on Goose Growth Performance, Body Size and Bare Skin Color

**DOI:** 10.3390/ani10081330

**Published:** 2020-08-01

**Authors:** Xiaoshuai Chen, Haiming Yang, Lei Xu, Xiaoli Wan, Zhiyue Wang

**Affiliations:** College of Animal Science and Technology, Yangzhou University, Yangzhou 225009, China; DX120170090@yzu.edu.cn (X.C.); hmyang@yzu.edu.cn (H.Y.); xlei@yzu.edu.cn (L.X.); wanxl@yzu.edu.cn (X.W.)

**Keywords:** broken rice, geese, growth performance, flipper score, bill score, serum biochemical variable

## Abstract

**Simple Summary:**

Geese are usually raised in the main rice-growing areas in China. Broken rice (BR) produced from rice processed into refined rice in these places can be applied to goose feed, which not only reduces the dependence on corn but can also reduce the transportation cost of feed raw materials by using local sources. It was found that BR had no negative effects on the growth performance of growing geese.

**Abstract:**

This study investigated the effect of replacing dietary corn with broken rice (BR) on goose growth performance, body size and bare skin color. In total, 240 28-day-old healthy male Yangzhou goslings with similar body weight (BW) were randomly divided into five groups, with six replicates per group and eight geese per replicate. The control group was fed with a corn-soybean meal. The BR_25_, BR_50_, BR_75_ and BR_100_ groups had 25%, 50%, 75% and 100% of corn replaced with BR, respectively (corresponding to 15.95%, 31.88%, 47.63% and 62.92% of BR in the feed, respectively), each with constant metabolizable energy (ME) to crude protein (CP) ratio (ME/CP). At 28, 42, 56 and 70 d, BW and feed intake for each pen were measured. Blood was collected, and body size and bare skin color were evaluated at 70 d. The results showed that different BR replacement proportions had no effect on BW at 42, 56 or 70 d or on average daily feed intake (ADFI) or average daily gain (ADG) from 28 to 42 d (*p >* 0.05) but BR_50_ and BR_75_ decreased the feed/gain ratio (F/G) from 28 to 42 d (*p* < 0.05). From 42 to 56 d, BR_75_ and BR_100_ geese had a lower ADFI than the control geese (*p* < 0.05), and BR_75_ and BR_100_ geese had a lower F/G than the BR_25_ geese (*p* < 0.05). Group BR_50_, BR_75_ and BR_100_ geese had a lower ADFI from 56 to 70 d than the control geese (*p* < 0.05). From 28 to 70 d, BR_50_, BR_75_ and BR_100_ groups had a lower ADFI (*p* < 0.05). Interestingly, the control and BR_25_ groups had a higher flipper score than the BR_50_, BR_75_ and BR_100_ groups (*p* < 0.05), and the control group had a higher flipper score than the BR_25_ group (*p* < 0.05). All BR groups reduced the bill scoring (*p* < 0.05). Different BR replacement proportions did not negatively affect serum biochemical variable at 70 d (*p >* 0.05). Overall, under these conditions, BR can totally replace corn in goose diets, and we recommend 75% replacement of corn with BR from 28 to 70 d.

## 1. Introduction

Corn is one of the main energy sources in poultry diets, providing 50–70% of the total energy of poultry. However, corn can also be used as human food and raw material for producing bioethanol. As China and other countries import an increasing amount of corn, the price of corn is also increasing. Under the trend of increasing global demand for corn, poultry producers are being forced to seek local raw materials to replace corn in animal feed. A large number of researchers have investigated the effect of different feed ingredients on the performance of poultry such as rice bran [1,2], defatted rice bran [3,4], paddy rice [5,6,7], whole wheat [8,9,10] and millet [11,12,13]. China is the largest rice producer in the world, accounting for 27.63% of global rice production [14]. When paddy is processed, broken rice (BR) with smaller particles is produced. As rice production and demand increase, the amount of BR produced worldwide also increases. When paddy rice is made into white rice, approximately 15% will be converted into BR or powder due to processes such as defusing and polishing [15]. However, BR is not preferred for human consumption, and it is disposed of for animal feeds because of its lower cost. Studies have shown that corn and rice have differences in starch type and starch particle size. Because rice contains higher amylopectin and smaller starch granules than corn, rice is more easily digested by animals than corn [16]. However, BR contains fewer xanthophylls and carotenoids than corn, which may affect the skin color of poultry when feeding BR. Skin color is a major concern among consumers in the marketplace, especially in China. Broken rice is widely used in pig production. Vicente et al. [17] showed that feeding BR improved dietary component digestibility and ileal morphology in piglets compared with those for corn. Amornthewaphat and Attamangkune [18] reported that pigs fed with BR had greater nutrient digestibility than those fed maize diets. Vicente et al. [19] and Mateos et al. [20] revealed that the inclusion of BR in the diet as a substitution for corn increased nutrient digestibility and growth performance in pig. However, the effect of BR on geese has not been studied yet.

BR replacing corn in feed totally or partly can not only reduce the dependence on corn in the breeding industry and save cost, but also alleviate the conflict between human and animals competing for food. We hypothesize that corn could be replaced by BR with no negative effect on goose performance. Therefore, the effects of various levels of corn replaced by BR on growth performance, serum biochemical variables, body size and bare skin color in geese were investigated in the present study.

## 2. Materials and Methods

### 2.1. Experimental Design and Diets

The Yangzhou University Animal Care and Use Committee approved all bird-handling protocols used in the study, with permit number SYXK (Su) IACUC 2012-0029.

A total of 240 28-d-old healthy male Yangzhou goslings from the same hatch were obtained from a commercial hatchery. The goslings were randomly assigned to five dietary treatments, with six replicate pens per treatment and eight geese per pen. Control group was fed with a corn-soybean meal basal diet; The BR_25_, BR_50_, BR_75_ and BR_100_ indicate that 25%, 50%, 75% and 100% of corn was replaced by broken rice in the basal diet, respectively (15.95%, 31.88%, 47.63% and 62.92% of BR, respectively, was added to the diets). All the groups had a constant metabolizable energy (ME) to crude protein (CP) ratio (ME/CP, 72.68 MJ/kg). In this experiment, a nipple drinker was used to provide drinking water for geese. There are windows around the goose house, so that sunlight can enter. In addition, the goose house has a wet curtain cooling system and a ventilation system to stabilize room temperature and humidity. Water and feed were provided ad libitum. The geese were exposed to natural daylight, and the room temperature was maintained at approximately 24 °C. The size of the replicate pen was 2.7 m^2^ (1.5 m × 1.8 m).

The chemical composition of the BR and corn in this study are shown in Table 1. The composition and nutrient levels of the experimental diets are shown in Table 2. The experimental diets were formulated mainly according to prior research results from our laboratory [21].

### 2.2. Sampling and Measurements

Growth performance was evaluated in terms of body weight (BW), average daily feed intake (ADFI), average daily gain (ADG) and feed-to-gain ratio (F/G). The BW of the birds in each pen was measured at 28, 42, 56 and 70 d of age. The ADFI, ADG and F/G values for the 28 to 42, 42 to 56 and 56 to 70 d periods were calculated. At 70 d, one goose from each pen with the average BW of the replicate was selected. No goose from any treatment died during the whole period of the experiment. Approximately 2.5 mL of blood was collected via the wing vein and centrifuged at 2000× *g* for 10 min at 4 °C to harvest serum, which was then stored at −20 °C until analysis of serum biochemical variable. The serum biochemical variable was determined using a UniCel Synchron D × C 800 fully automatic biochemical analysis system (Beckman Coulter, Los Angeles, CA, USA). The serum biochemical variable was performed following the instructions of the corresponding commercial kits (Nanjing Jiancheng Bioengineering Institute, Nanjing, China). Serum total protein (TP) was assayed by the Biuret method. Serum albumin (ALB) was assayed by the Microplate Assay method. Serum glucose (GLU) was assayed by the GLU Oxidase method. Serum uric acid (UA), cholesterol (CHO) and triglycerides (TGs) were assayed by the Enzyme-colorimetric method. Serum high-density lipoprotein (HDL) and low-density lipoprotein (LDL) were assayed by the Direct method.

After blood collection was completed, each goose was evaluated for body measurement, and flipper and bill color were given a score. The body size were conducted using a measuring tape calibrated in centimeters (cm) for half-diving length (the distance from the tip of the bill to the midpoint of the hip line), keel length (the distance between the anterior and posterior tips of the keel), shank length (the linear distance from the upper joint of shank to the third and fourth toes) and shank circumference (the circumference of the middle tibia). Measurements were carried out using the method described by the Agricultural Ministry of China (NY/T823-2004) [22]. We selected five geese, which represented five levels of flipper and bill color grades. Standards of color grading were set up with scores from 1 to 5 points corresponding to colors from white to yellow. Then, the flippers and bills were compared among all geese according to the standards of Color Grading.

The dry matter (DM) was determined by drying the samples in a drying oven (DHG-9240A, Shanghai, China). The crude protein (CP) was determined by the Kjeldahl method. The crude fat (EE), calcium (Ca) and total phosphorus were analyzed according to the standard procedures set forth by the Association of Official Analytical Chemists (AOAC, 2005) [23]. The amino acids (AA) were analyzed by amino acid analyzer.

### 2.3. Statistical Analyses

All the data were initially processed using Excel and analyzed using a one-way ANOVA procedure in SPSS 19.0 (SPSS, 2010) [24]. Significant differences among the treatment means were determined at *p* < 0.05 by Duncan’s multiple range tests.

## 3. Results

### 3.1. Growth Performance

The effects of BR on the BW of geese at 28, 42, 56 and 70 d are shown in Table 3. The ADFI, ADG and F/G of geese fed with dietary BR are shown in Table 4. No significant differences were found in BW at 42, 56 and 70 d (*p >* 0.05). ADFI and ADG were also not affected from 28 to 42 d (*p >* 0.05), but BR_50_ and BR_75_ groups had a lower F/G from 28 to 42 d (*p* < 0.05). From 42 to 56 d, geese in the BR_75_ and BR_100_ groups had a lower ADFI than those in the control group (*p* < 0.05) and geese in the BR_75_ and BR_100_ groups had a lower F/G than those in the BR_25_ group (*p* < 0.05). Geese in the BR_50_, BR_75_ and BR_100_ groups had a lower ADFI from 56 to 70 d than those in the control group (*p* < 0.05). From 28 to 70 d, BR_50_, BR_75_ and BR_100_ groups had a lower ADFI (*p* < 0.05).

### 3.2. Body Size and Serum Biochemical Variable

The body size (cm) of geese at 70 d of age are shown in Table 5. The concentrations of dietary BR had no effect on half-diving length, keel length, shank length or shank circumference (*p >* 0.05).

The effects of BR on the serum biochemical variable of the geese at 70 d are shown in Table 6. The dietary BR had no significant effect on serum concentration of TP, ALB, GLO, A/G, BUN, UA, GLU, TG, HDL or LDL of geese at 70 d (*p >* 0.05).

### 3.3. Bare Skin Color

A one-way ANOVA showed that there were effects of dietary BR concentrations on bare skin color score of the flipper (*p* < 0.01) and bill (*p* < 0.01) (Figure 1). The higher the color score is, the more yellow the bare skin color; otherwise, the paler the color. The control and BR_25_ groups had a higher flipper score than the BR_50_, BR_75_ and BR_100_ groups (*p* < 0.01); and the control group had a higher flipper score than the BR_25_ group (*p* < 0.05). Broken rice had a significant effect on bill scoring (*p* < 0.01).

The flipper and bill scoring were carried out by selecting bird of average BW from each pen. The higher the score, the yellower the flippers and bills. Statistical analysis was carried out with one-way ANOVA followed by Duncan’s multiple range tests. The data are shown as means ± SEM (standard errors of the means, *n* = 6). Values within a group of columns with no common letters differ significantly (*p* < 0.05). Control group was fed with a corn-soybean meal basal diet; BR_25_, BR_50_, BR_75_ and BR_100_ indicate that 25%, 50%, 75% and 100% of corn was replaced by broken rice in the basal diet (15.95%, 31.88%, 47.63% and 62.92% of BR, respectively, was added to the diets).

## 4. Discussion

### 4.1. Growth Performance

Under the conditions of our experiment, no significant difference was found in BW at 42, 56 and 70 d. No difference was observed in ADFI and ADG from 28 to 42 d. Similar results were reported in meat-type quails by Filgueira et al. [25], who reported that meat-type quails fed with diets that replaced corn with BR did not differ in weight gain or feed intake from 7 to 49 d of age. These results indicated that replacing corn with BR in goose diets did not negatively affect BW (from 25% to 100%). In some major rice-producing areas, corn can be totally replaced with BR in goose diets. In our study, BR_50_ and BR_75_ reduced the F/G from 28 to 42 d. The reason may be that rice contains smaller starch granules (3 to 8 vs. 2 to 30 μm) [16], lower amounts of nonstarch polysaccharide [26], less amylose and higher ME and CP than corn. Kermanshahi et al. [27] demonstrated that NSP added to broiler chicken diets reduced the FCR. The mechanism was explained by Smits and Annison [28], who stated that soluble NSP can (1) cover absorption sites in the intestine and prevent nutrient absorption and (2) be consumed by harmful microbiota in the intestine and detrimentally affect host health. In addition, starch with a high amylose content is often less digestible. However, compared with the control group, the BR_100_ exhibited no significant effect on the F/G from 28 to 70 d. The reasons for this result are not clear, but it may be related to the ratio of amylopectin and amylose. This may be related to the metabolizable energy in diet. It was reported that feed intake can be significantly reduced in broilers fed with diets with high metabolizable energy [29]. Geese in the BR_75_ and BR_100_ groups had a lower ADFI than those in the control group; moreover, geese in the BR_75_ and BR_100_ groups had a lower F/G than those in the BR_25_ group from 42 to 56 d. We agreed with Gonzalez-Alvarado et al. [30], who studied the effects of different cereals on 1- to 21-d-old broilers. Their results showed that broilers fed with corn had a poorer F/G than broilers fed with BR (1.37 vs. 1.32). Interestingly, BR had a significant effect on ADFI but not on the F/G from 56 to 70 d of age. Geese in the BR_50_, BR_75_ and BR_100_ groups had a lower ADFI from 56 to 70 d than those in the control group. These results indicated that the best proportion of BR used instead of corn in the goose diet was 75% from 28 to 70 d.

### 4.2. Body Size and Serum Biochemical Variable

Body-size and serum biochemical variable can reflect the growth performance and health status in geese to a certain extent. Compared with the control geese, geese in the BR groups exhibited no significant effect on body size or serum biochemical variable in geese at 70 d. These results were consistent with the BW result at 70 d. However, there is no literature on the effect of BR on body size and serum biochemical variable of geese. The results provide evidence that replacing corn with BR in the diet had no negative effect of geese at 70 d.

### 4.3. Bare Skin Color

Interestingly, we found significant differences in bare skin color among the treatments. Color scores from 1 to 5 points correspond to colors from pale to yellow. The flipper score was significantly decreased in the BR_50_, BR_75_ and BR_100_ groups (31.88% to 62.92% of BR in feed) compared with the flipper score in the control and BR_25_ groups (0% and 15.95% of BR in feed), whereas the bill skin color turned paler and paler as the BR concentration increased from 0% to 100%. The results of the present study provide evidence that BR in the diet has a significant effect on the bare skin color of geese, especially on the color of bill. However, the effect of BR on the bare skin color of poultry has not yet been reported. Liu et al. [31] reported that adding okra to feed can be used successfully to provide pigment in broiler diets due to its substantial content of xanthophylls and carotenoids. Similarly, lutein diester [32] and marigold [33] can also provide pigment in broiler diets. Poultry cannot synthesize xanthophylls or carotenoids and must obtain these compounds from their diets. These pigments are then transported in the blood to the subcutaneous fat tissues and skin, where they are stored. Our results indicated that BR can reduce the bare skin quality in geese. Further research is needed to study the method to improve bare skin color in geese using pigments in the goose diet with BR.

## 5. Conclusions

In conclusion, replacing corn with BR in goose diets had no adverse effect on the BW of geese from 28 to 70 d. Moreover, 47.63% of BR in feed decreased the F/G. However, the addition of BR reduced the bare skin color on geese at 70 d. Under our experimental conditions, BR can completely replace corn in goose diets, and we recommend that 75% of BR be used instead of corn in goose diets from 28 to 70 d.

## Figures and Tables

**Figure 1 animals-10-01330-f001:**
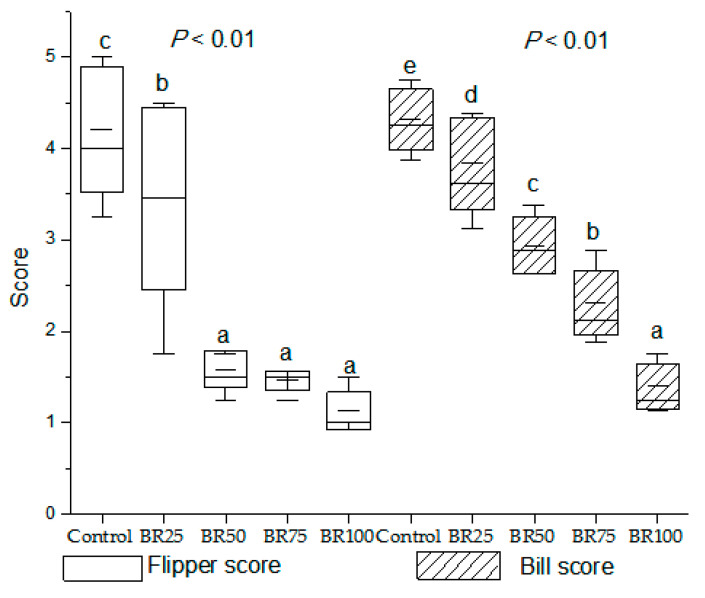
Effects of broken rice on flipper and bill score of geese at 70 days of age.

**Table 1 animals-10-01330-t001:** Analyzed nutrient content of broken rice (BR) and corn in this study (%).

Items ^1^	BR	Corn	Items	BR	Corn
DM	87.59	87.04	Methionine	0.15	0.10
CP	8.13	7.91	Arginine	0.65	0.35
EE	1.79	5.35	Threonine	0.21	0.21
Ash	0.36	1.28	Histidine	0.18	0.17
Ca	0.01	0.02	Isoleucine	0.31	0.21
TP	0.07	0.21	Leucine	0.70	0.83
AA			Phenylalanine	0.41	0.31
Lysine	0.27	0.22	Valine	0.40	0.27

^1^ DM, dry matter; CP, crude protein; EE, crude fat; Ca, calcium; TP, total phosphorus; AA, amino acid.

**Table 2 animals-10-01330-t002:** Composition and nutrient levels of experimental diets for days 28 to 70 (as-fed basis).

Item	Groups ^1^				
	Control	BR_25_	BR_50_	BR_75_	BR_100_
Ingredients (%)					
Corn	65.00	47.86	31.88	15.88	0.00
Broken rice	0.00	15.95	31.88	47.63	62.92
Soybean meal	23.18	22.26	21.61	20.92	20.17
Rice bran	0.00	2.11	2.67	3.49	4.75
Limestone	0.89	1.01	1.04	1.11	1.17
Calcium hydrogen phosphate	1.09	0.93	0.87	0.78	0.68
Salt	0.30	0.30	0.30	0.30	0.30
DL-Methionine	0.20	0.19	0.18	0.17	0.17
Lysine	0.07	0.05	0.04	0.02	0.00
Premix ^2^	1.00	1.00	1.00	1.00	1.00
Rice husk	8.27	8.34	8.54	8.71	8.85
Total	100.00	100.00	100.00	100.00	100.00
Nutrient composition ^3^					
ME (MJ/kg)	11.29	11.42	11.57	11.71	11.84
CP (%)	15.53	15.72	15.92	16.11	16.29
ME/CP (MJ/kg) ^4^	72.68	72.68	72.68	72.68	72.68
Crude fiber (%)	6.06	6.06	6.06	6.06	6.06
Lysine (%)	0.84	0.84	0.84	0.84	0.84
Methionine (%)	0.43	0.43	0.43	0.43	0.43
Total phosphorus (%)	0.58	0.58	0.58	0.58	0.58
Calcium (%)	0.77	0.77	0.77	0.77	0.77

^1^ Control group was fed with a corn-soybean meal basal diet; The BR_25_, BR_50_, BR_75_ and BR_100_ indicate that 25%, 50%, 75% and 100% of corn was replaced by broken rice in the basal diet, respectively (15.95%, 31.88%, 47.63% and 62.92% of BR, respectively, was added to the diets). ^2^ One kilogram of premix contained Vitamin A, 1,200,000 IU; Vitamin D, 400,000 IU; Vitamin E, 1800 IU; Vitamin K, 150 mg; Vitamin B_1_, 60 mg; Vitamin B_2_, 600 mg; Vitamin B_6_, 200 mg; Vitamin B_12_, 1 mg; nicotinic acid, 3 g; pantothenic acid, 900 mg; folic acid, 50 mg; biotin, 4 mg; choline, 35 mg; Fe (as ferrous sulfate), 6 g; Cu (as copper sulfate),1 g; Mn (as manganese sulfate), 9.5 g; Zn (as zinc sulfate), 9 g; I (as potassium iodide), 50 mg; Se (as sodium selenite), 30 mg. ^3^ Calculated values. ^4^ ME/CP, The metabolizable energy: protein ratio.

**Table 3 animals-10-01330-t003:** Effects of broken rice on body weight (g) in geese from 28 to 70 days of age ^1^.

Age	Groups ^2^					SEM ^3^	*p*-Value
	Control	BR_25_	BR_50_	BR_75_	BR_100_		
28	1139	1138	1138	1138	1139	2.22	1.00
42	2344	2385	2395	2364	2365	34.94	0.88
56	3503	3458	3477	3418	3483	42.04	0.73
70	4162	4055	4025	4020	4068	57.19	0.45

^1^ Each value represents the mean of six replicates. ^2^ Control group was fed with a corn-soybean meal basal diet; BR_25_, BR_50_, BR_75_ and BR_100_ indicate that 25%, 50%, 75% and 100% of corn was replaced by broken rice in the basal diet, respectively (15.95%, 31.88%, 47.63% and 62.92% of BR, respectively, was added to the diets). ^3^ Standard error of mean.

**Table 4 animals-10-01330-t004:** Effects of broken rice on the average daily feed intake (g), average daily gain (g) and the ratio of feed to gain of geese from 28 to 70 days of age ^1^.

Days of Age	Item ^2^	Groups ^3^					SEM ^4^	*p*-Value
		Control	BR_25_	BR_50_	BR_75_	BR_100_		
28–42	ADFI	220	218	216	205	218	5.87	0.44
	ADG	86.1	89.0	89.8	87.6	87.6	2.51	0.87
	F/G	2.56 ^c^	2.45 ^abc^	2.41 ^ab^	2.34 ^a^	2.49 ^bc^	0.05	0.03
42–56	ADFI	285 ^c^	277 ^bc^	265 ^abc^	245 ^a^	262 ^ab^	6.67	<0.01
	ADG	82.8	76.7	77.3	75.3	79.9	2.24	0.21
	F/G	3.47 ^ab^	3.62 ^b^	3.44 ^ab^	3.25 ^a^	3.28 ^a^	0.07	0.03
56–70	ADFI	313 ^b^	300 ^ab^	273 ^a^	277 ^a^	281 ^a^	8.54	0.01
	ADG	47.0	42.6	39.2	43.0	41.8	3.07	0.53
	F/G	6.78	7.21	7.05	6.60	7.02	0.48	0.91
28–70	ADFI	267 ^c^	260 ^bc^	248 ^ab^	237 ^a^	249 ^ab^	5.15	<0.01
	ADG	72.0	69.4	68.7	68.6	69.8	1.36	0.45
	F/G	3.72	3.74	3.61	3.46	3.57	0.07	0.08

^a–c^ Values within a row with no common letters differ significantly (*p* < 0.05). ^1^ Each value represents the mean of six replicates. ^2^ ADFI, average daily feed intake; ADG, average daily gain; F/G, the ratio of feed to gain. ^3^ Control group was fed with a corn-soybean meal basal diet; the BR_25_, BR_50_, BR_75_ and BR_100_ indicate that 25%, 50%, 75% and 100% of corn was replaced by broken rice in the basal diet, respectively (15.95%, 31.88%, 47.63% and 62.92% of BR, respectively, was added to the diets). ^4^ Standard error of mean.

**Table 5 animals-10-01330-t005:** Effects of broken rice on body size (cm) in geese at 70 days of age ^1^.

Items	Groups ^2^					SEM ^3^	*p*-Value
	Control	BR_25_	BR_50_	BR_75_	BR_100_		
Half-diving length	72.08	72.25	71.25	72.17	71.50	1.05	0.96
Keel length	15.42	15.58	15.92	15.58	16.50	0.40	0.38
Shank length	12.55	12.25	12.15	12.62	12.00	0.42	0.85
Shank circumference	5.77	5.83	6.10	6.17	6.13	0.12	0.11

^1^ Each value represents the mean of six replicates. ^2^ Control group was fed with a corn-soybean meal basal diet; BR_25_, BR_50_, BR_75_ and BR_100_ indicate that 25%, 50%, 75% and 100% of corn was replaced by broken rice in the basal diet, respectively (15.95%, 31.88%, 47.63% and 62.92% of BR, respectively, was added to the diets). ^3^ Standard error of mean.

**Table 6 animals-10-01330-t006:** Effects of broken rice on serum biochemical variables of geese at 70 days of age ^1^.

Items ^2^	Groups ^3^					SEM ^4^	*p*-Value
	Control	BR_25_	BR_50_	BR_75_	BR_100_		
TP (g/L)	47.57	45.02	45.40	45.56	48.11	1.80	0.68
ALB (g/L)	13.92	13.62	13.62	13.08	14.48	0.40	0.23
GLO (g/L)	33.65	31.40	31.78	32.48	33.63	1.48	0.76
A/G	0.42	0.44	0.43	0.41	0.43	0.01	0.43
UA (mmol/L)	195	241	206	251	249	22.49	0.30
GLU (mmol/L)	10.75	10.28	10.64	9.46	10.93	0.46	0.58
CHO (mmol/L)	4.32	4.44	4.39	4.48	5.06	0.25	0.30
TG (mmol/L)	0.51	0.46	0.53	0.49	0.64	0.08	0.55
HDL (mmol/L)	1.95	2.13	2.12	2.02	2.39	0.15	0.37
LDL (mmol/L)	1.61	1.51	1.51	1.62	1.79	0.10	0.36

^1^ Each value represents the mean of six replicates. ^2^ TP, total protein; ALB, albumin; GLO, globulin; UA, uric acid; GLU, glucose; CHO, cholesterol; TG, triglycerides; HDL, high-density lipoprotein; LDL, low-density lipoprotein. ^3^ Control group was fed with a corn-soybean meal basal diet; BR_25_, BR_50_, BR_75_ and BR_100_ indicate that 25%, 50%, 75% and 100% of corn was replaced by broken rice in the basal diet, respectively (15.95%, 31.88%, 47.63%, and 62.92% of BR, respectively, was added to the diets). ^4^ Standard error of mean.

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
