# Peer review of "Effect of Replacing Dietary Corn with Broken Rice on Goose Growth Performance, Body Size and Bare Skin Color"

_animals, 2020, doi:10.3390/ani10081330_

Round 1

Reviewer 1 Report

This study investigated effect of replacement of corn with broken rice on growth performance, blood variables and skin color in duck. The study is not highly original as there are many studies replacing corn or other ingredients with rice. However, study is useful. The statistics should be revised as stated under comments.

The experiment design was sound.

The number of replicates is reasonable.

The data are sound.

The manuscript is well written.

I have few comments below.

L30-31: Overall treatment F-test was not significant for the shank circumstances. You should not use MCT when overall F test is not is significant. 

L44-46: There are also many studies on broken rice. You should cite a few of them as you have cited a lot for rice bran for 6 references. Delete of them and include broken rice also as it is one of your main ingredients.

L83: One reference is sufficient.

Table 1: why were ME and CP greater in the BR diets while you changed a number of ingredients. You should have kept these values as low as possible at least when you have changes all the ingredients inn the diets. It indicates you are giving favour for the BR diets.

Table 1: Did you analyse at least few major chemical composition?

L94-95: did you adjusted the mortality for intake or FCR, etc?

L110-111: How did you measure skin color? Was any instrument used? How did you consider 5 point and other as 1 point? Need more description here.

L114-115: You should use LSD when F-test is significant. Moreover, why did you use LSD which gives more type I error.

Table 2: no decimal places are needed for mean and SEM.

Table 2: Item should be replaced with age of duck.

Table 3: No decimal places is needed when mean is >100. 

Table 3: When P>0.05 you should not use superscript as you stated in the stat section you used LSD when P<0.05.

Table 4: When P>0.05 you should not use superscript. You should also revise the texts.

L148: font size should be similar.

L156-163: It seems it  should be in one paragraph.

L159: what was F-test P-value.

L165-172: They should be in one paragraph. Present F-test P value.

L192-193: your diet had alos greater ME and CP for the BR diet.

L225-230: There are several papers to increase skin color and meat color such as leaves, flower petals, etc. You should cite few of them. Okra is not common to use.

L237-244: You should state about skin color also. It is consumer's requirement to have good skin and meat color.

Author Response

Thank you very much for your advice. I have learned a lot from it.

Reviewer 2 Report

Reviewer’s comments on “animals-816942” tilted “Effect of replacing dietary corn with broken rice on 2 goose growth performance, body-size measurements 3 and bare skin color”

General comments: The manuscript covers the field of much needed research i.e. “alternatives to energy feedstuffs for cost effective and viable animal feed manufacturing” however the paper lacks many critical details and thus cannot be accepted for publication in the current format. Hopefully, the specific comments below will help to improve the manuscript.

Specific Comments:

Abstract:

Line 3: Use of term “high level” is ambiguous. Revise sentence.

Line 21: ME should be “metabolisable energy”

Line 25: Be consistent with your use of terminology. “31.88% and 47.63% BR should be replaced with the abbreviations BR50, BR75.

Line 30: You have written “ Compared to the control geese, BR75 geese had a greater shank 30 circumference (P < 0.05).” That is not correct. According to Table 4, the P-Value for Shank circumference is 0.11 which means that differences are non-significant.

Line 33: “BR had a significant effect on beak scoring (P < 0.05)”. What kind of effect? Positive or negative?? Not clear?

Introduction:

Line 43: Producer ? who? Poultry producer of feed producers?

Line 48 and 62: Sentence needs revision (correct tense should be used)

Materials and Methods:

This section needs to be rewritten as many details are missing.  Why too many fibre sources were added in the diet (rice bran, rice husk)? Is it normal to have such a high fibre diet for Geese?

Line 80:  Details missing? How were they exposed to day light? What was lighting schedule? Must mention, what type of housing was provided? What type of bedding material was used? What was the temperature and humidity during the experimental period? Where Geese kept indoor throughout the experiment or they had outdoor access?  What was the size of replicate pen? Not clear why experimental period started at day 28? How were birds reared in pre-trial period?

Line 94: Where geese weighed individually, or bulk weighed?

Line 110-111: Method of body size measurement and bare skin colour scoring must be provided as the reference mentioned is not available online.

Results:

Were experimental diets analysed to determine actual proximate analysis? What was the actual fibre content of the diets? 

Discussion

Line 189-190: The statement “These results indicated that BR replaced corn in goose diets without negatively affecting BW or feed intake” is not correct. Inclusion of BR resulted in around 11% reduction in feed intake. Revise the sentence.

Line 194: “Kermanshahi 193 et al. [31] demonstrated that NSP added to broiler chicken diets increased the F/G.” How? Elaborate?? Must mention the reason given by Kermanshahi for this change. Also, not sure what do you mean by increased F/G? Do you mean improved? Good or bad?

Line 197: “The reasons for this result are not clear, but it may be related to the ratio of amylopectin and amylose”? How? What is the ideal ratio? What was the ratio of amylopectin and amylose in your study?

Line 213: The statement “The geese in the BR75 group had a greater shank circumference than those in the control group” is not correct? P value is 0.11.

The discussion section should elaborate on the fact that in the current study feed intake is significantly reduced but there is no effect on body weight or weight gain but has in fact improved F/G ratio.? How? Must speculate why?

Table 4: The statement “a-b Values within a row with no common letters differ significantly (P < 0.05)” has to be removed as none of the values are significant. Remove superscripts from Shank circumference as P= 0.11

Author Response

(The authors gave the same response as above.)

Reviewer 3 Report

The paper could be interesting, but for some parameters more details are needed for a right discussion of many results.

Some Tables do not show units, and they show (typing?) mistakes.

row 10: are raised: usually? in the past?

row 13: no negative: what does it mean? for growing birds, until slaughter age?

row 21: metabolic?

row 30: a greater shank circumference. Any explanation? also in the text. Is it a positive result? Previous results on slaughter performance should be given in the text.

row 62: references?

Table 1. air-dry basis? Indications on BR % and replaced corn (groups) ,as indicated in the text and tables, do not allow an easy reading and understanding of the text.

Is it correct the premix composition?

2 the metabolic energy:protein ratio...??

3 calculated values: chemical composition of broken rice (and other rice components of the diet) should be given.

row 96: why 1 goose/replicate....? is it correct from a statistical point of view? for evaluating the studied parameters?

row 97: more details on blood sampling are needed for the results  and discussion on the blood profile. Furthermore, you should indicate in the discussion the choise and importance of these blood parameters (and not others)  for the health status.

row 111: more details on the range of 5-point system.

Table 2 and Others: why 2 decimals when high values (thousands or hundreds) are showed ?

table 3: units! furthermore, check the a, b, c order!

table 4: letters in last row!

table 5: units!

row 188: year!

row 191: if a widely diffused feeding (rice), why a research, now? no previous results and indications?

row 215: why ..beneficial to tibia development? which could be the diet composition effect?

row 230: any analysis on BR or rice components on this topic?

Conclusions: is it positive to have a lower skin yellowness index ? and a higher shank circumference? which is the slaughter age of these birds?

Author Response

(The authors gave the same response as above.)

Reviewer 4 Report

The topic of the manustript is very important from the econamicaly point of view especially for the producers not only geese but also chicken broilers. Authors concluded that replacing corn with broken rice in goose diets had no adverse effect on the BW or serum chemistry of geese from 28 to 70 d. BR diet decreased ADFI.

Comments:

"Slaughter performance is one of the important indicators of the economic benefits of breeding and also a key indicator of the growth performance of meat animals".

Probably the Authors made slaughter analysis. If its possible please add results concerning: slaughter yield, semi-eviscerated carcass yield, eviscerated carcass yield, breast yield, thigh yield, abdominal fat yield. If you will reduce the size of the Figure 1 and 2 it will be enough space to put a table with slaughter performance.

Minor comments:

What about mortality during the experiment?

If the mean value of given trait did not differ signifficantly between groups the Authors don't need to indicate it with (P>0.05).

Could you specify detection kits that the authors have used in serum chemistry evaluation?

Discussion is rather poor in references. I realize that there are no may papers about broken rice in poultry but the discussion doesn’t look well. The Authors have just repeated the results.

Table 2- check superscripts

Author Response

(The authors gave the same response as above.)

Round 2

Reviewer 1 Report

The manuscript has been improved. I have following comments:

Comment: Table 1: Why were ME and CP greater in the BR diets while you changed a number of ingredients. You should have kept these values as low as possible at least when you have changes all the ingredients inn the diets. It indicates you are giving favour for the BR diets.

Answer: The diet composition is designed according to the same metabolisable energy: protein ratio.According to the Table of Feed Composition and Nutritive Value in China, the protein content and metabolic energy of broken rice are higher than that of corn, so it is difficult to design the nutrition level of each group to the same level.When designed the formula, broken rice was not favored.

Re-comment: Authors should discuss this aspects as high ME value is supposed to decrease feed intake in poultry, it was not due to broken rice itself. 

Comment: Table 1: Did you analyse at least few major chemical composition?

Answer: So far, the protein, fat, ash and moisture contents of diets, broken rice, corn, soybean meal and rice bran have been determined.The contents of energy, calcium, phosphorus and crude fiber have not been determined yet.

Recomment: This is a nutritional study. Authors must report the analysed composition of some of the major composition. I find Authors have reported calculated values in the table 1. What was the CP, fat, ash content of the broken rice. What was the ME and CP values as per the Table of Feed Composition and Nutritive Value in China. Provide the reference?

Comments: L94-95: did you adjusted the mortality for intake or FCR, etc?

Answer: Goose has not died since 28 days old.

Authors should state it in the manuscripts.

Other comments

L79: is to was

L111-114: The to the when it is in the middle of a sentence.

L116-117: Revise English.

L118-119: Delete the sentence. It will be in the statistical section.

L149: circumference (P > 0.05).

Figure 1 and 2 can be combined in a figure with two panels.

L236-237: Cite reference.

Author Response

Comments and Suggestions for Authors

The manuscript has been improved. I have following comments:

Comment: Table 1: Why were ME and CP greater in the BR diets while you changed a number of ingredients. You should have kept these values as low as possible at least when you have changes all the ingredients inn the diets. It indicates you are giving favour for the BR diets.

Answer: The diet composition is designed according to the same metabolisable energy: protein ratio. According to the Table of Feed Composition and Nutritive Value in China, the protein content and metabolisable energy of broken rice are higher than that of corn, so it is difficult to design the nutrition level of each group to the same level. When designed the formula, broken rice was not favored.

Re-comment: Authors should discuss this aspects as high ME value is supposed to decrease feed intake in poultry, it was not due to broken rice itself.

Answer: This point has been added in the discussion and references have been cited.

Comment: Table 1: Did you analyse at least few major chemical composition?

Answer: So far, the protein, fat, ash and moisture contents of diets, broken rice, corn, soybean meal and rice bran have been determined. The contents of energy, calcium, phosphorus and crude fiber have not been determined yet.

Recomment: This is a nutritional study. Authors must report the analysed composition of some of the major composition. I find Authors have reported calculated values in the table 1. What was the CP, fat, ash content of the broken rice. What was the ME and CP values as per the Table of Feed Composition and Nutritive Value in China. Provide the reference?

Answer: The chemical composition of the broken rice and corn in this study has been added in the manuscript.

 Comments: L94-95: did you adjusted the mortality for intake or FCR, etc?

Answer: Goose has not died since 28 days old.

Authors should state it in the manuscripts.

Answer: It has been revised.

 Other comments

L79: is to was

Answer: It has been revised.

L111-114: The to the when it is in the middle of a sentence.

Answer: It has been revised.

L116-117: Revise English.

Answer: The language has been reorganized and modified.

L118-119: Delete the sentence. It will be in the statistical section.

Answer:  It has been deleted

L149: circumference (P > 0.05).

Answer: It has been revised.

Figure 1 and 2 can be combined in a figure with two panels.

Answer:  It has been combined in a figure with two panels.

L236-237: Cite reference.

Answer: It has been revised.

Reviewer 2 Report

Reviewers further minor comments on manuscript Title: Effect of replacing dietary corn with broken rice on goose growth performance, body-size measurements and bare skin color

Authors: XiaoShuai Chen, Haiming Yang, Lei Xu, Xiaoli Wan, Zhiyue Wang *

The manuscript has been improved. However, the methodology section still needs improvements so that reader can understand rearing conditions.  The following comments may help to further improve the manuscript before it can be published.

Specific comments:

Line 46: Replace “will be” with “is”

Line 52: Replace “In addition” with “However”

Line 79: Details still missing. Must include the details in the manuscript. You have provided the following information to the reviewer:

“There are windows around the goose house, through windows sunlight can enter the goose house. The geese were exposed to natural daylight. The goose house itself has a lighting system and windows. In addition, the goose house also has a wet curtain cooling system and a ventilation system. Such a goose house can ensure proper temperature and good ventilation. The room temperature was maintained at approximately 24°C, and room humidity was maintained at approximately 70%. Geese kept indoor throughout the experiment. The experiment geese were bought from 28 days old. The reason for starting from 28 days old is that 28 days old is the brooding period of geese, and the intake of food during brooding is relatively low.”

The same information that is provided to the reviewer can be summarised and included in the manuscript. The reader needs to know that geese were reared indoors throughout the experimental period, how the temperature and cooling was maintained and that the experimental period started at the end of the brooder phase.

Also mention if there was any bedding material used in pens and if there was any water pond inside the pen?

Line 116-17: Sentence needs revision. Grammar to be checked and corrected.

Line 117: “color will be rated as 1 to 5 point from light to deep”. Where shades of yellow or brown were used? Still not clear? For example, to measure yolk colour, Roche yolk fan is used, which has standard shades of yellow and each shade has a number.  So what was used to compare flippers and beak? Still not clear???

Line 192-194.: Revise sentence. Simply say no differences were observed (P>0.05).

Line 195: “BR replaced corn in goose diets without negatively affecting BW “ Must mention what was the level of BR used in the study referred.

Line 237: Moever? Do you mean Moreover?? Please check.

Line 250: “of geese” Replace “of” with “on”

Author Response

The manuscript has been improved. However, the methodology section still needs improvements so that reader can understand rearing conditions.  The following comments may help to further improve the manuscript before it can be published.

Specific comments:

Line 46: Replace “will be” with “is”

Answer: It has been revised.

Line 52: Replace “In addition” with “However”

Answer: It has been revised.

Line 79: Details still missing. Must include the details in the manuscript. You have provided the following information to the reviewer:

“There are windows around the goose house, through windows sunlight can enter the goose house. The geese were exposed to natural daylight. The goose house itself has a lighting system and windows. In addition, the goose house also has a wet curtain cooling system and a ventilation system. Such a goose house can ensure proper temperature and good ventilation. The room temperature was maintained at approximately 24°C, and room humidity was maintained at approximately 70%. Geese kept indoor throughout the experiment. The experiment geese were bought from 28 days old. The reason for starting from 28 days old is that 28 days old is the brooding period of geese, and the intake of food during brooding is relatively low.”

The same information that is provided to the reviewer can be summarised and included in the manuscript. The reader needs to know that geese were reared indoors throughout the experimental period, how the temperature and cooling was maintained and that the experimental period started at the end of the brooder phase.

Also mention if there was any bedding material used in pens and if there was any water pond inside the pen?

Answer: These points have been written in the article. this experiment, geese used nipple type drinker, so there was no water pond inside the pen.

Line 116-17: Sentence needs revision. Grammar to be checked and corrected.

Answer: The language has been reorganized and modified.

Line 117: “color will be rated as 1 to 5 point from light to deep”. Where shades of yellow or brown were used? Still not clear? For example, to measure yolk colour, Roche yolk fan is used, which has standard shades of yellow and each shade has a number.  So what was used to compare flippers and beak? Still not clear???

Answer: Our laboratory has a Roche yolk fan for measuring the color of egg yolk, but after comparison, we find that the Roche yolk fan for measuring the color of egg yolk can not evaluate the color of flippers and beaks of geese. Therefore, we selected several flippers and beaks with obvious color differences as the standard to evaluate the color of flippers and beaks. The picture below shows the flippers and beaks that we selected as the color evaluation standard.

Because the color will change after beak separation, I took more photos and then chose the obvious difference as the scoring standard

1 point                                      2 point

3 point                                      4 point

5 point                                        

Line 192-194.: Revise sentence. Simply say no differences were observed (P>0.05).

Answer: It has been revised.

Line 195: “BR replaced corn in goose diets without negatively affecting BW “Must mention what was the level of BR used in the study referred.

Answer: It has been added.

Line 237: Moever? Do you mean Moreover?? Please check.

Answer: I’m so sorry, I misspelled the word and I had revised it in the manuscript.

Line 250: “of geese” Replace “of” with “on”

Answer: It has been revised.

I'm really sorry because of these mistakes. I found that you reviewed my manuscript so carefully, which made me realize the rigor of scientific research and the responsibility of work. Thank you.

Reviewer 3 Report

The paper has been improved, but the current paper shows:

mistakes (typing?) in the text, tables and figures, no or poor discussion on some points (blood profile, xantophylls content, ...) (as requested).

Furthermore, the statistical approach for some parameters (colour scoring, ..) is inadequate, both for the constitution of the samples and for the kind of test used. 

The current paper is not available for publication.

Author Response

The paper has been improved, but the current paper shows:

mistakes (typing?) in the text, tables and figures, no or poor discussion on some points (blood profile, xantophylls content, ...) (as requested).

Furthermore, the statistical approach for some parameters (colour scoring, ..) is inadequate, both for the constitution of the samples and for the kind of test used.

The current paper is not available for publication.

Answer: I added the nutrient content of broken rice and corn in the manuscript, and some wrong sentences have been revised. In addition, for the color score, we tried to use the Roche yolk fan when measuring, but the Roche yolk fan of egg yolk is not suitable for the color score of flippers and beaks. However, the color of beak and flippers of geese are obviously different. Therefore, we selected five flippers and beaks with significant differences as the scoring criteria. Because the color will change after beak separation, I took more photos. The picture below shows the flippers and beaks that we selected as the color evaluation standard. I hope my manuscript is easy to understand.

1 point                                      2 point

3 point                                      4 point

5 point      

In addition, it is very meaningful to use broken rice instead of corn, and few people are doing it now. Therefore, I hope my experiment can provide a reference for everyone.

This manuscript is a resubmission of an earlier submission. The following is a list of the peer review reports and author responses from that submission.